# Design of Two Compact Wideband Monopoles Through Loading with Linear Approximated Lumped Components

**DOI:** 10.3390/mi15121477

**Published:** 2024-12-07

**Authors:** Jiansen Ma, Weiping Cao, Xinhua Yu

**Affiliations:** Guangxi Key Laboratory of Wireless Wideband Communication and Signal Processing, Guilin University of Electronic Technology, Guilin 541004, China; majiansen@foxmail.com (J.M.); yusilian@126.com (X.Y.)

**Keywords:** load antenna, wideband monopoles, colinear structure

## Abstract

In this paper, two ultra-wideband monopoles in a colinear structure are presented for application in remote terrestrial communication systems. The antennas consist of a loaded monopole with a hat and an elevated loaded monopole located in the upper position. All lumped loads are modeled as linear frequency-dependent components to approximate the practical component property for achieving ultra-wideband characteristics, since the constant value property of a component is only present in a relatively narrow band. The antennas are simulated by the method of moments (MoM) with asymptotic waveform evaluation (AWE) to speed up frequency sweep across a wide bandwidth. For proper simulation with the AWE process, the parallel RLC load with linear frequency-dependent components is modeled in a corresponding impedance function. With the optimized load parameters, one antenna covers 30–750 MHz with a VSWR < 3.5 and the other one covers 800 MHz–3000 MHz with a VSWR < 2.5, which are promising results for terrestrial omnidirectional applications.

## 1. Introduction

Monopole antennas, with their omnidirectional characteristics, are widely used in many wireless communication systems, such as broadcasting [1,2], vehicle communication [3,4,5], indoor local communication [6,7], and electromagnetic sensors [8,9,10]. With the development of multifunctionality on a platform, more and more wireless communication systems are assembled on a single platform [11,12]. This leads to many antennas working for different systems being placed nearby, which can result in unwanted electromagnetic interference problems [13,14]. Even if each system is well designed with satisfactory isolation, the use of too many antennas increases the complexity on a platform. Therefore, an omnidirectional antenna covering all system bands is promising, as it means that only one antenna needs to be considered on the platform. In addition, as the demand for information capacity grows, the bandwidth of wireless communication systems is designed to be wider over the years [15,16,17,18]. Wideband or multiband antennas are always a hot research area as they have the potential to meet the wideband requirements of wireless communication systems.

As a basic concept, the ordinary quarter-wavelength monopole is usually equivalent to a quarter-wavelength open-circuit lossless transmission line. It is a resonant structure, and the current on it conforms to a standing wave distribution [19]. The quarter-wavelength monopole has satisfactory gain in all directions and a good voltage standing wave ratio (VSWR) only in a narrow bandwidth. An increase in frequency causes a reverse current, which results in gain degradation in all directions according to the superposition theorem [20]. On the other hand, when the frequency decreases, the monopole becomes electrically small, and its equivalent circuit is gradually turned toward an open circuit [21]. It is hard to feed an electrically small antenna with a 50 ohm port. Therefore, there is a natural bandwidth limitation of the ordinary quarter-wavelength monopole, and bandwidth expansion of the monopole has always been a popular research topic in various monopole applications.

To expand the bandwidth of the monopole, one popular method is to add a hat at the free end of the monopole, which is equivalent to adding a capacitor between the open circuit and ground. A displacement current can exist strongly in the low-impedance path of the equivalent capacitor, and a traveling wave distributes on the monopole, which could result in a wide bandwidth characteristic. With printed circuit board (PCB) technology, the hat load of the monopole can be designed in various shapes and is compact in microwave applications [22,23]. Some novel structures with special functions can also be embedded in the monopole with a hat. Ref. [24] presented a printed monopole antenna with a modified stepped impedance resonator (SIR) in the hat load to achieve dual-band operation. The size of the monopole can be reduced by introducing an embedded resonant structure [25]. To expand the bandwidth further, other techniques such as a sleeve structure can be employed at the bottom of the monopole [26]. Since physical size is not a primary concern in microwave applications, it is quite flexible to design antennas in various structures using PCB technology.

In VHF/UHF applications, the size of the monopole antenna increases considerably, especially on the lower side of the band. The classical wire monopole is always a popular candidate compared with other structures and is widely used in omnidirectional applications. Moreover, as the outline of the wire monopole is small, the wind resistance of the antenna can be effectively reduced in outdoor applications [27]. To expand the bandwidth of the wire monopole while maintaining its outline, loading with lumped elements is feasible as long as a suitable loading strategy is employed. Loading the monopole with only resistance is the simplest way to try to expand the bandwidth of the antenna. Since the loaded resistance would absorb part of the power on the antenna, a traveling wave is established on the antenna, which would result in an expansion of the bandwidth [28,29]. Inductance or capacitance loading is also used to alter the reverse current on the monopole to obtain a wideband characteristic [30,31,32]. To alter the reverse current effectively, a parallel RLC circuit is fairly suitable since the current would vary intensely when the circuit is in resonance. Moreover, the resistance in the circuit can absorb power to some extent, which might benefit the port characteristics of the antenna. Refs. [33,34,35,36] present many strategies to determine the position and element value of the parallel RLC circuit. However, all of these discussions are based on an assumption that all element values of the parallel RLC circuit are constant, which might fail in the ultra-wideband range. To realize the required wideband characteristic, tedious tuning on the prototype is unavoidable.

In this paper, two monopoles in a compact structure are presented with ultra-wideband characteristics through loading with a parallel RLC circuit. All elements are modeled as linear frequency-dependent elements to model practical component behaviors. To simulate the antenna across an ultra-wide bandwidth, the antenna is simulated using the method of moments (MoM) with an asymptotic waveform evaluation (AWE) sweeping process for efficiency compared with sweeping discrete points one by one across such a wide band. The parallel RLC circuit with linear frequency-dependent elements is further modeled in the AWE loading model. The lower monopole also loads a hat in spokes to obtain a good port match at the lowest side of the concerned band. Moreover, this hat also acts as an elevated ground for the upper monopole. The outer conductor of the coaxial feed line for the upper monopole is designed to act as a loaded inductor for the lower monopole since this feed line has to travel through the lower monopole. In addition, this feeding line with a ferrite bead functions as part of an attenuator for the lower monopole. With these shared structure designs, the two monopoles are designed in a compact dual-band radiation system. The positions and element values of the parallel RLC circuit are determined by genetic algorithm (GA) optimization. Then, the optimized antenna is verified through experiments.

## 2. Antenna Design

### 2.1. Parallel Lumped Load of Monopole

Compared with ideal lumped components, practical lumped components behave differently across a wide bandwidth, especially coil inductors. Since the parasitic capacitance between each turn of a coil inductor is not negligible, this capacitance needs to be considered carefully in wideband applications. With the parasitic capacitance, a coil inductor would act as a parallel RC circuit and might resonate in a wide bandwidth, as shown in Figure 1. The impedance of a coil inductor increases more rapidly with respect to frequency. In contrast to a constant-value element, the effective inductance $L_{effective}$ as a function of frequency can be deduced from the classical impedance function of an inductor.
(1)$$Z=j\omega L_{effective}$$
A coil inductor with N = 3, *p* = 0, D = 1 cm, and d= 1.5 cm has an effective inductance of 30–800 MHz, as Figure 2 illustrates. Before resonating, the coil inductor still exhibits inductance and varies with frequency.

As Figure 2 depicts, the inductance of the coil inductor could be further simplified to a linear frequency-dependent approximation as follows:(2)$$L_{appro}(k)=b_{L}\cdot (A_{L}\cdot k+B_{L})$$
with
(3)$$A_{L}=\frac{M_{L}-1}{f_{2}-f_{1}}\cdot \frac{c_{0}}{2\pi }$$
(4)$$B_{L}=1-\frac{M_{L}-1}{f_{2}-f_{1}}\cdot f_{1}$$
where $L_{appro}$ as a function of the wave number k is the approximate inductance of the concerned band, $f_{1}$ and $f_{2}$ are the lower and upper frequency of the concerned band, $M_{L}$ is the ratio of the inductance at the upper frequency to the inductance at the lower frequency, $c_{0}$ is the velocity of light, and $b_{L}$ is the inductance at the lower frequency. In addition, a similar approximation could also apply to the practical patch resistance and capacitance as follows:(5)$$R_{appro}(k)=b_{R}\cdot (A_{R}\cdot k+B_{R})$$
(6)$$C_{appro}(k)=b_{C}\cdot (A_{C}\cdot k+B_{C})$$
where $R_{appro}$ and $C_{appro}$ are the approximated resistance and capacitance of the practical patch resistor and capacitor, respectively. $b_{R}$, $A_{R}$, $B_{R}$, $b_{C}$, $A_{C}$, and $B_{C}$ are the corresponding coefficients of linear frequency-dependent models, similar to the approximation of the inductor. Even though the resistance of the practical patch resistor and the capacitance of the practical capacitor would not vary intensively like the coil inductor in a wide bandwidth, alleviation could be made for the tuning process since there would be multiple loads arranged on the monopole.

The impedance of the loaded parallel lumped RLC circuit as a function of wave number can be expressed as follows:(7)$$Z(k)=\frac{1}{1/R+1/\left(jc_{0}kL\right)+jc_{0}kC}$$
Substituting (2), (5), and (6) into (7) gives the following:(8)$$Z(k)=\frac{P_{3}k^{3}+P_{2}k^{2}+P_{1}k^{1}+P_{0}}{Q_{5}k^{5}++Q_{4}k^{4}+Q_{3}k^{3}+Q_{2}k^{2}+Q_{1}k+Q_{0}}$$
with
(9)$$P_{3}=jc_{0}A_{R}A_{L}\cdot b_{R}b_{L}$$
(10)$$P_{2}=jc_{0}(A_{R}B_{L}+B_{R}A_{L})\cdot b_{R}b_{L}$$
(11)$$P_{1}=jc_{0}B_{R}B_{L}\cdot b_{R}b_{L}$$
(12)$$P_{0}=0$$
(13)$$Q_{5}=-c_{0}^{2}A_{R}A_{L}A_{C}\cdot b_{R}b_{L}b_{C}$$
(14)$$Q_{4}=-c_{0}^{2}(A_{R}A_{C}B_{L}+A_{L}A_{C}B_{R}+A_{R}A_{L}B_{C})\cdot b_{R}b_{L}b_{C}$$
(15)$$Q_{3}=-c_{0}^{2}(A_{R}B_{L}B_{C}+A_{L}B_{R}B_{C}+A_{C}B_{R}B_{L})\cdot b_{R}b_{L}b_{C}$$
(16)$$Q_{2}=-c_{0}^{2}B_{R}B_{L}B_{C}\cdot b_{R}b_{L}b_{C}+jc_{0}A_{L}\cdot b_{L}$$
(17)$$Q_{1}=A_{R}\cdot b_{R}+jc_{0}B_{L}\cdot b_{L}$$
(18)$$Q_{0}=B_{R}\cdot b_{R}$$

To obtain an accurate simulation result across a wide bandwidth for the loaded antenna, adequate sweep points should be set in the concerned wide band range. Since too many frequency points have to be covered, sweeping these points discretely by the method of moments (MoM) would be time-consuming. Employing the asymptotic waveform evaluation (AWE) procedure could accelerate the sweeping process in the simulation, since a bandwidth ratio (BWR) of over 2:1 could be covered by AWE [37]. In contrast to just loading the ideal constant value of a lumped component in the impedance matrix of the MoM, obtaining the nth derivative of the impedance of the loaded parallel lumped RLC circuit as a function of wave number is necessary. To comply with this requirement, the rational polynomial form of the impedance of the loaded parallel lumped RLC circuit (8) is expanded into a partial fraction presentation for derivation as follows:(19)$$Z(k)=\sum_{i=1}^{Nz}\frac{r_{i}}{k-q_{i}}$$
where $r_{i}$ is the residue of (8), and $q_{i}$ is the corresponding pole. And the nth derivative of (19) can be derived as follows:(20)$$Z^{\left(n\right)}(k)=\sum_{i=1}^{N_{z}}{\left(-1\right)}^{n}\cdot n!\cdot \frac{r_{i}}{{\left(k-q_{i}\right)}^{n+1}}$$

Now, it is possible to proceed with the diagonal loading of the impedance matrix in the AWE process using (20). And then electromagnetic solving follows the normal AWE process to obtain the electromagnetic property of the loaded monopole in a wide bandwidth for optimization.

### 2.2. Design of Colinear Loaded Monopoles

As shown in Figure 3, two monopoles are designed in a colinear structure with a parallel lumped RLC circuit loaded dispersedly on the vertical copper tube of the antenna. First of all, the lower monopole is designed to cover a 30–750 MHz band. The length of the lower monopole is *L*_1_ = 175 cm. And a cross blade with a length *L*_3_ = 15 cm is arranged at the top working as a hat load for obtaining a good match at the lower side of the concerned band. Seven parallel lumped RLC circuits are loaded on the lower monopole for obtaining acceptable gain in all directions across the concerned band while lowering the VSWR to some extent. Since it is hard to obtain a transmission line transformer working for the 30–750 MHz band, an attenuator is attached to the port of the lower monopole for lowering the VSWR further to an acceptable extent. All element values and the position of these parallel lumped RLC circuit loads are determined by optimization later.

The upper monopole is designed to cover the 800–3000 MHz band and its length is *L*_2_ = 15 cm. Since *L*_2_ ≥ *L*_3_ is satisfied, the cross blade also works as an elevated ground for this upper monopole. Three parallel lumped RLC circuits are loaded on the lower monopole for obtaining acceptable gain in all directions across the concerned band while lowering the VSWR to some extent. This upper monopole is fed by a coaxial line traveling through the inside of the tube of the lower monopole, as depicted in Figure 4. In this case, the outer surface of the outer conductor of the coaxial feeding line and the internal surface of the tube would form another coaxial waveguide. To remove this waveguide effect, the coaxial feeding line is attached to the internal surface of the tube so that the outer surface of the outer conductor of the coaxial feeding line and the tube are an equipotential body now. Moreover, at the load position, the coaxial feeding line is wound so that the outer surface of the outer conductor of the coaxial feeding line could be designed as a coil inductor. Combined with the patch resistor and the patch capacitor, there is a completely parallel lumped RLC circuit load unit arranged at the load position, as shown in Figure 4.

As the outer surface of the outer conductor of the coaxial feeding line is designed to be identical to the radiator of the lower monopole, special consideration should be given to the port of the lower monopole. As depicted in Figure 5, some ferrite beads are arranged on the coaxial feeding line as a choke to feed the current to the lower monopole effectively, since the outer surface of the outer conductor of the coaxial feeding line is commonly connected to the ground at the connector of the upper monopole. Moreover, as shown in Figure 5, combined with two other patch resistors, a π-type circuit with attenuation characteristics is designed for the attenuator of the lower monopole, as depicted in Figure 3. By adjusting this two resistors’ values carefully, it is easy to obtain the required attenuation characteristics. Then, this attenuator can be connected to a 50 ohm connector for the lower monopole by a 50 ohm microstrip transmission line, as shown in Figure 3.

### 2.3. Optimization of Loaded Monopoles

Both loaded monopoles are optimized by GA optimization following a similar process to [33,34,35,36] but using the loading technique described in Section 2.2 for accurately modeling practical load behaviors. To accurately perform simulations across a wide bandwidth, the frequency band was not sampled uniformly, but very densely at lower frequencies, viz. every 0.5 MHz from 30 to 40 MHz, every 1 MHz from 40 to 70 MHz, every 2 MHz from 70 to 120 MHz, every 5 MHz from 120 to 300 MHz, every 10 MHz from 300 to 600 MHz, every 20 MHz from 600 to 750 MHz and from 800 to 1500 MHz, and every 50 MHz from 1500 to 3000 MHz. Nonuniform sampling is important not only for increasing the relative weight of the gain in the lower portion of the frequency band, but also for eliminating possible narrowband dips in gain which have a tendency to appear at a lower frequency. Combining the loading technique described in Section 2.2 in the MoM solution with the AWE process, solution sweeping across an ultra-wide band with massive points could be carried out efficiently. The loaded elements for optimization are $b_{R}$, $b_{L}$, and $b_{C}$ in (2), (5), and (6). The lower and upper bounds of these load parameter values were set to 0 < R < 2000 ohm, 0 < L < 400 nH, and 0 < C < 100 pF for the lower monopole and 0 < R < 2000 ohm, 0 < L < 25 nH, and 0 < C < 10 pF for the upper monopole. All of the component values are represented by 8 bits in GA optimization. Hence, the location of each load for the lower monopole and the upper monopole is specified by 6 bits and 4 bits, respectively, in GA optimization. The goal of optimization was satisfied when both antennas were optimized at the same time. The optimized value of the loaded elements as well as the locations of the loads are presented in Table 1 for the lower monopole and in Table 2 for the upper monopole. The location of the load circuits corresponds to their distances to the feed point.

The optimized gain of the lower monopole is shown in Figure 6, and the optimized gain is above −4 dB below 200 MHz and above 0 dB above 200 MHz. For comparison, the gain of an unloaded monopole with the same dimension is also depicted in Figure 6. The optimized loaded monopole avoids gain dips in all directions that appear naturally in the unloaded monopole due to pattern split. Figure 6 also plots the gain of the loaded monopole with the same load parameters in Table 1 but with a constant value. In the case of constant load, the gain deviates from the expected level far away across a wide band. However, the gain of the loaded monopole with the component of the approximate model appears well in the concerned band. Figure 7 shows the optimized gain of the upper monopole, and the optimized gain is above 0 dB from 800 to 3000 MHz. For comparison, the gain of an unloaded monopole with the same dimension is also depicted in Figure 7, and the dip of the gain in the concerned band is also removed in the load version. As shown in Figure 7, the constant load also deviates from the expected level, even though the BWR of the upper monopole is not great compared to the lower monopole. Both loaded monopoles sharing some common structures are optimized with satisfactory gain in all directions at the same time. Figure 8a–i and Figure 9a–c show the simulated radiation patterns of the lower monopole and the upper monopole, respectively, at a typical frequency in the concerned band. Although the radiation pattern of these two loaded monopoles splits as the frequency increases in the concerned band, the gains of these two loaded monopoles in all directions have not declined with the help of the loads.

## 3. Experiments and Results

The optimized loaded monopole is fabricated as Figure 10a,b show. A copper tube is divided into eight segments producing seven gaps for arranging seven load circuits for the lower monopole. And the yellow plastic tube is employed as a bracket for keeping the linear structure of the lower loaded monopole. Since the material of this plastic tube is identical to an ordinary radome, there is little effect on the electric property of the antenna. The load unit of the lower monopole is composed of patch resistance, patch capacitance, and the coil inductor made of the outer surface of the outer conductor of the coaxial feeding line, as Figure 10c shows, which corresponds to the design depicted in Figure 4. The attenuator with 1.5 dB attenuation for the lower monopole is shown in Figure 10d. Ferrite beads choke the current to the ground while working as part of the attenuator at the same time, as Figure 5 depicts.

The experiment is carried out in outdoor open half-space environment, as shown in Figure 11. The transmitting and the receiving antenna are separated 100 m away for measurement of the far field at 30 MHz by employing the criterion of 10λ for far-field judgment. The measured and simulated VSWR of the lower monopole are shown in Figure 12a. It can be seen that the lower monopole covers a wide band of 30–750 MHz with a VSWR < 3.5. Acceptable agreement between the simulation and measurement can be found. The differences between the simulation and measurement may come from the fact that the practical load circuit possesses a certain square in contrast to the point load in the simulation. The measured and simulated gain of the lower monopole are shown in Figure 12b. Since it is unavoidable that multiple interferences appear in outdoor measurements, the measured gain vibrates slightly across the band. However, there is still acceptable agreement between the simulation and measurement. For the upper monopole, the plots of the measured and simulated VSWR and the measured and simulated gain are shown in Figure 13a,b, respectively. Since the feed line of the upper monopole traveled through the lower tube over 2 m, 2.5 dB attenuation is effectively introduced on the port of the upper monopole, which results in a better VSWR (<2), while the gain has not deteriorated yet. The introduced attenuator and loaded resistor absorb some power feed on the antenna port for matching across a wide bandwidth. The radiation efficiency of the lower monopole and the upper monopole shown in Figure 14 is acceptable in ultra-wide-band loaded monopole applications.

## 4. Performance Comparison

To further illustrate the advantages of the presented colinear loaded monopoles, the comparisons with other loaded monopoles are listed in Table 3. Refs. [33,35,38] also design monopoles using the load technique but employ a constant loaded component in their simulation which might require more effort in tuning the prototype. And these works have a similar gain and VSWR in only one frequency band. Ref. [39] proposed a design with a dual wide band but their minimum gain is lower compared with other works. Ref. [40] achieved a better gain and VSWR with a dual band but its BWR is lower. Our proposed antenna has two frequency bands with a compact size, and both monopoles have a wider band with an acceptable gain and VSWR.

## 5. Conclusions

Two novel compact lumped–loaded monopoles achieving wide-band characters are presented in this paper. The two monopoles are designed to share a common structure and are compact in size. By modeling all lumped components of the load in the approximated model, the optimized characters of the loaded antenna are more accurately simulated across a wide bandwidth, which is verified by the measured result of the prototype. The proposed design is suggested for terrestrial communication systems and has possible applications in emergency omnidirectional searching systems.

## Figures and Tables

**Figure 1 micromachines-15-01477-f001:**
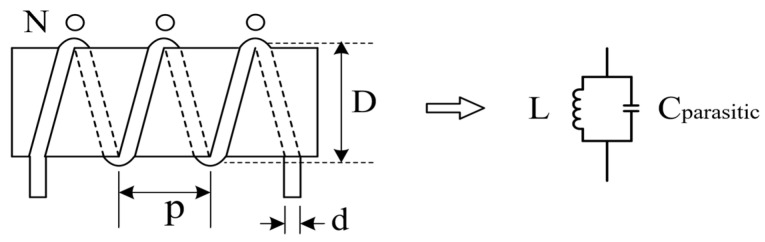
Coil inductor with parasitic capacitor behaves as parallel resonant circuit.

**Figure 2 micromachines-15-01477-f002:**
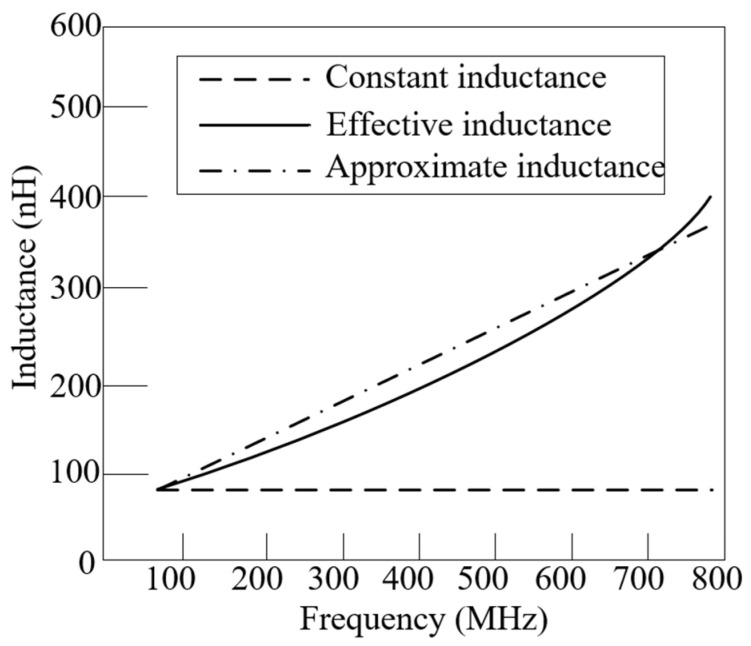
Inductance of coil inductor varies versus frequency in wide bandwidth.

**Figure 3 micromachines-15-01477-f003:**
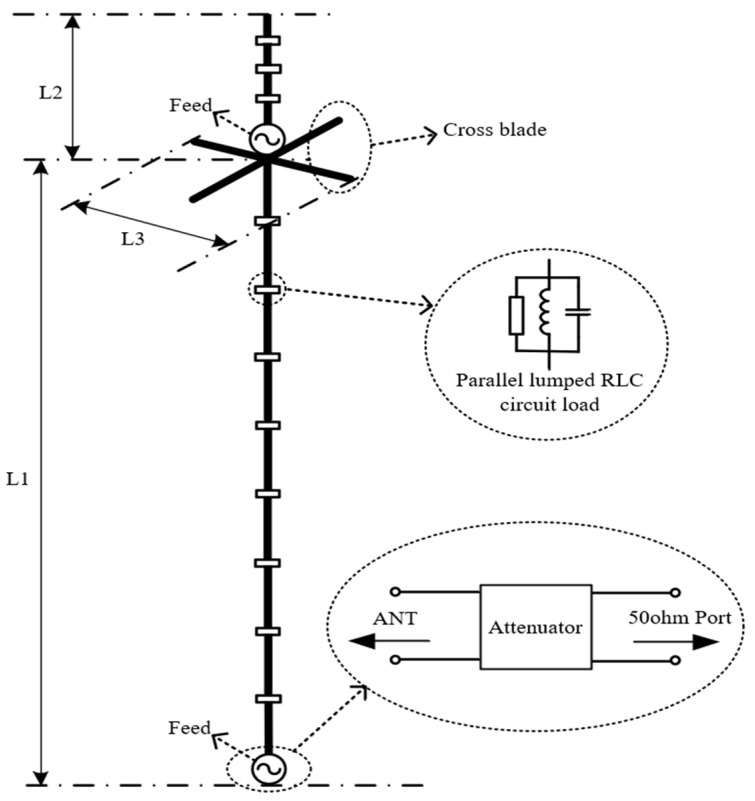
Parallel lumped RLC loaded monopoles arranged in colinear structure.

**Figure 4 micromachines-15-01477-f004:**
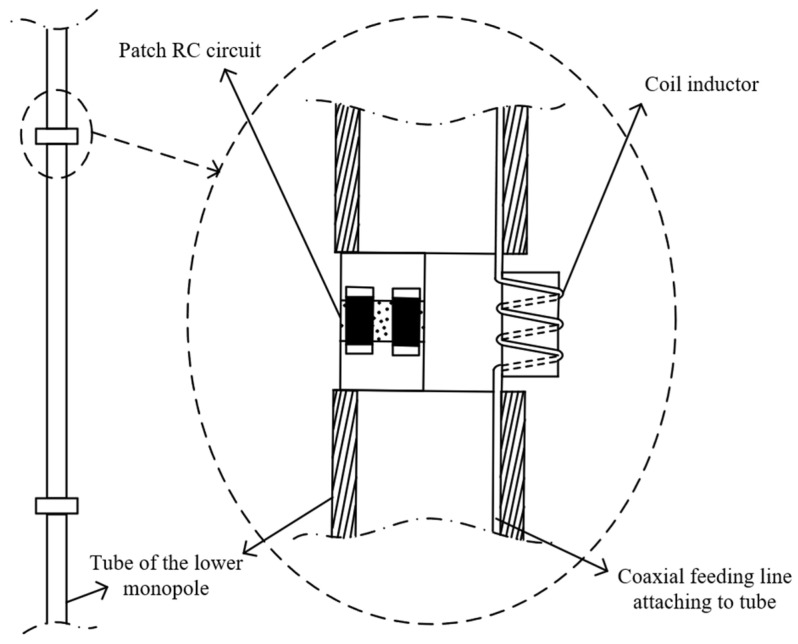
The coaxial line for feeding the upper monopole travels through the inside of the tube of the lower monopole. The figure shows the practical load design for the lower monopole.

**Figure 5 micromachines-15-01477-f005:**
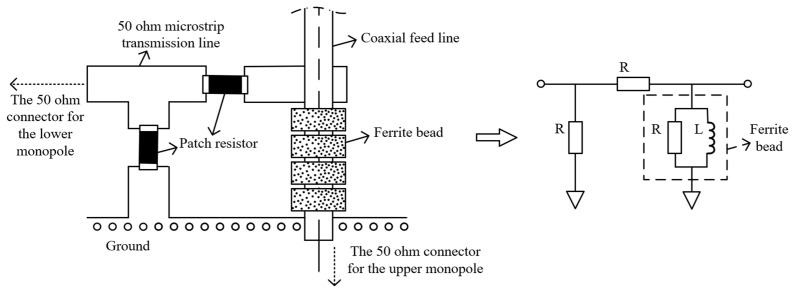
Attenuation of lower monopole by π-type resistive circuit composed partly of ferrite beads.

**Figure 6 micromachines-15-01477-f006:**
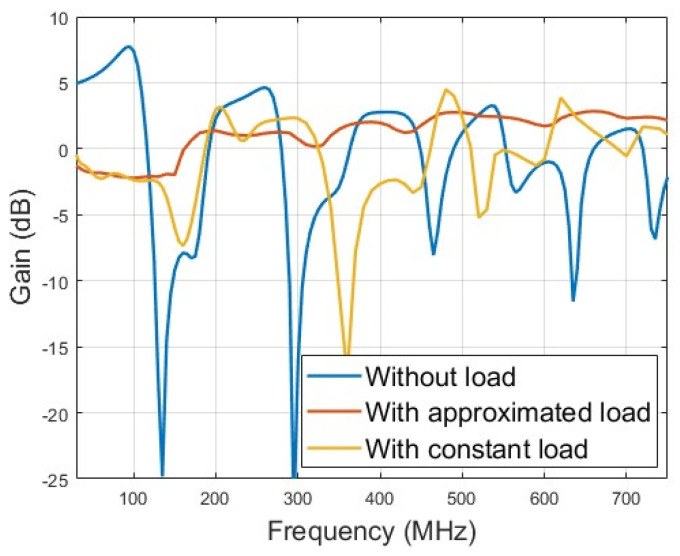
Optimized gain of lower monopole compared with directivity of unloaded monopole with same dimension and loaded monopole with constant value component.

**Figure 7 micromachines-15-01477-f007:**
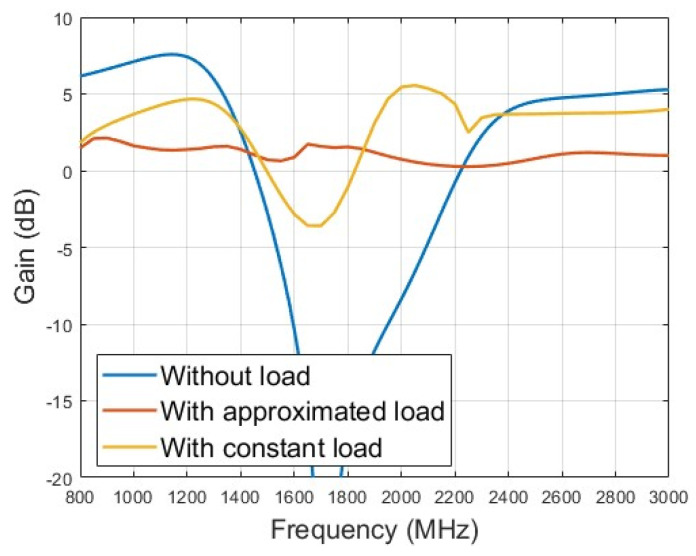
Optimized gain of upper monopole compared with directivity of unloaded monopole with same dimension and loaded monopole with constant value component.

**Figure 8 micromachines-15-01477-f008:**
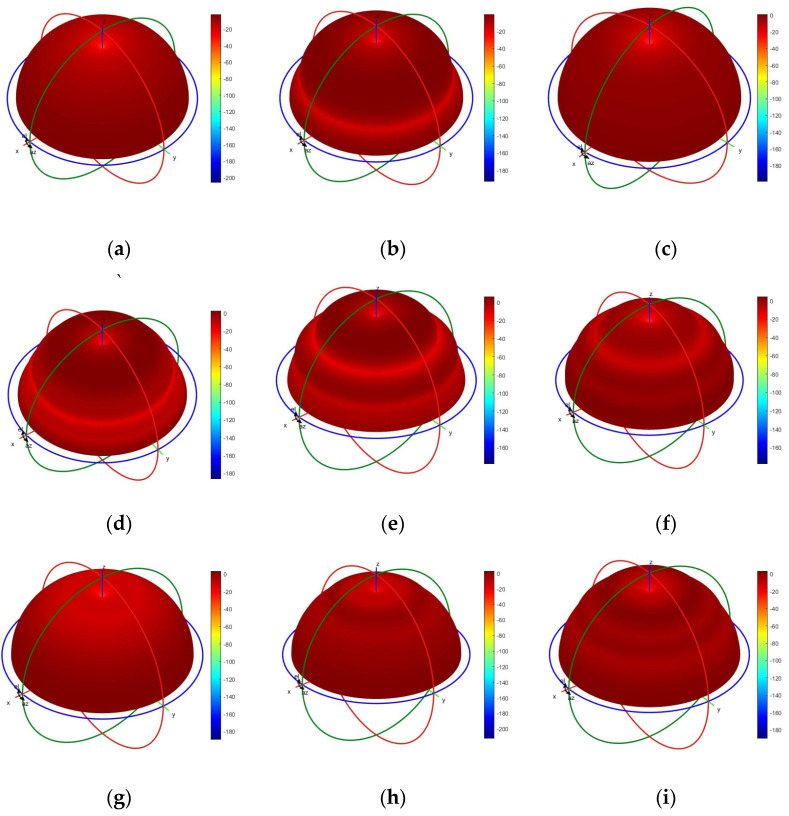
The simulated radiation patterns of the lower monopole. (**a**) At 75 MHz. (**b**) At 150 MHz. (**c**) At 225 MHz. (**d**) At 300 MHz. (**e**) At 375 MHz. (**f**) At 450 MHz. (**g**) At 525 MHz. (**h**) At 600 MHz. (**i**) At 675 MHz.

**Figure 9 micromachines-15-01477-f009:**
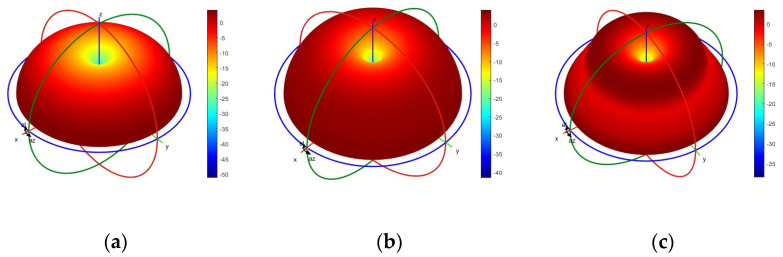
The simulated radiation patterns of the upper monopole. (**a**) At 1000 MHz. (**b**) At 1800 MHz. (**c**) At 2600 MHz.

**Figure 10 micromachines-15-01477-f010:**
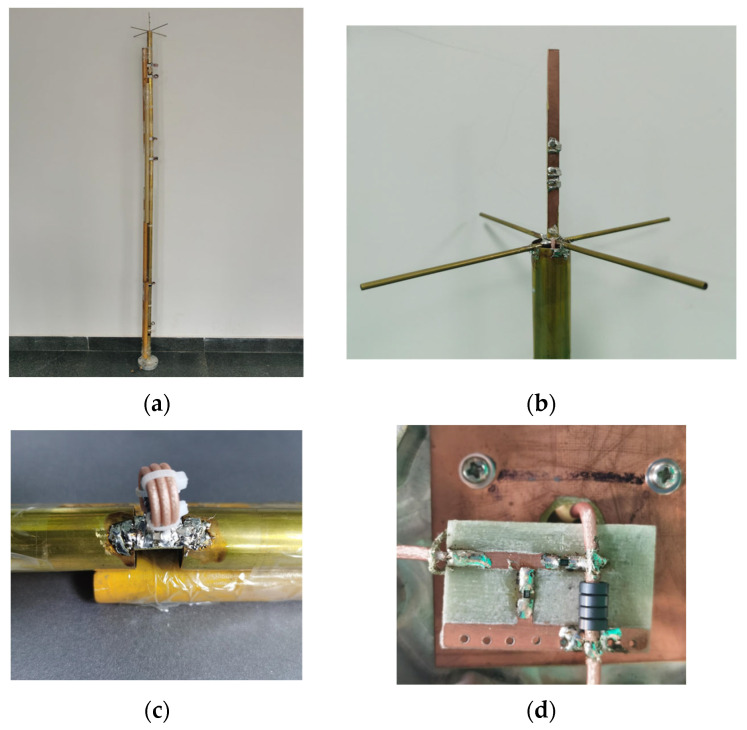
Fabricated prototypes. (**a**) Two loaded monopoles in colinear structure; (**b**) upper monopole; (**c**) unit of load circuit; (**d**) attenuator for lower monopole.

**Figure 11 micromachines-15-01477-f011:**
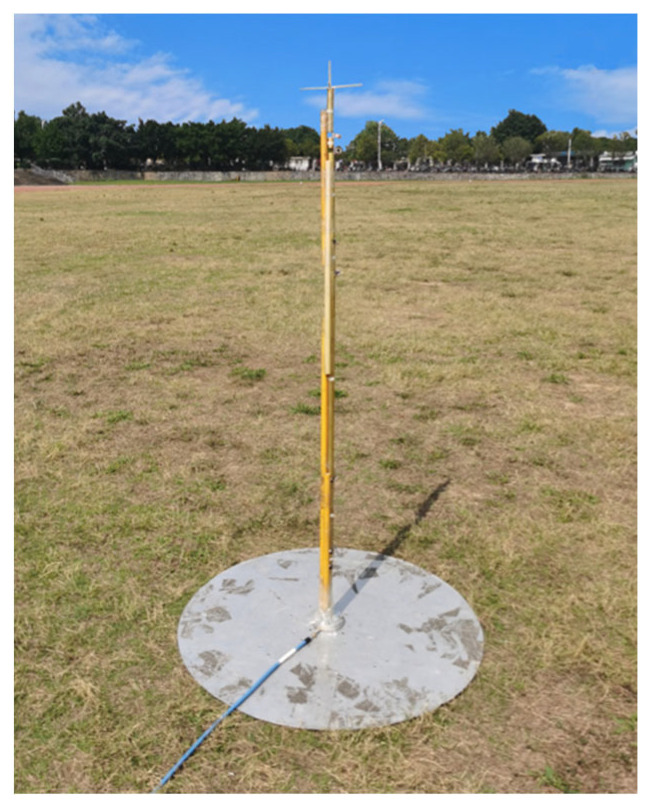
Experiment in outdoor open half-space environment.

**Figure 12 micromachines-15-01477-f012:**
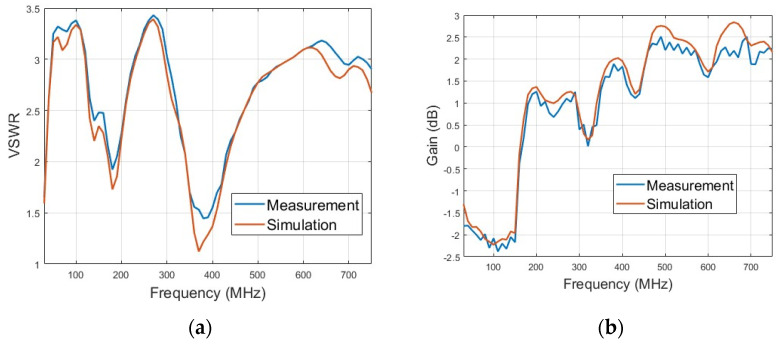
Electrical property of lower monopole. (**a**) Measured and simulated VSWRs. (**b**) Measured and simulated gains.

**Figure 13 micromachines-15-01477-f013:**
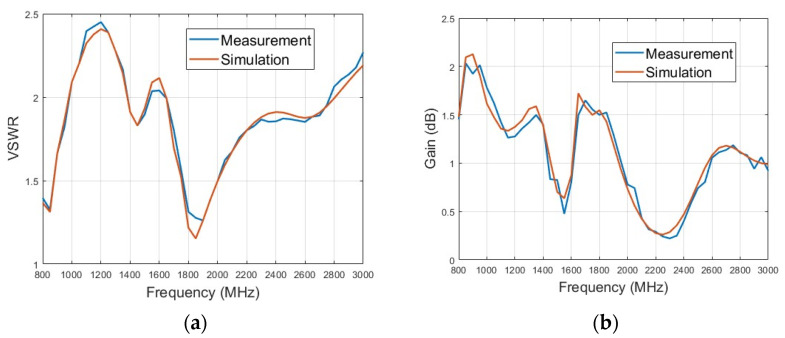
Electrical property of upper monopole. (**a**) Measured and simulated VSWRs. (**b**) Measured and simulated gains.

**Figure 14 micromachines-15-01477-f014:**
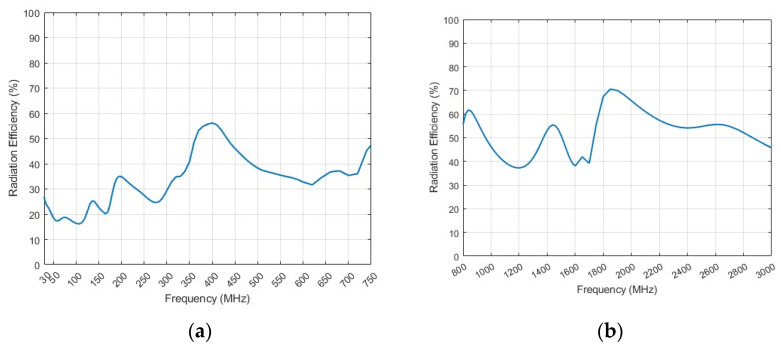
(**a**) Radiation efficiency of lower monopole. (**b**) Radiation efficiency of upper monopole.

**Table 1 micromachines-15-01477-t001:** Parameters of optimized load of lower monopole.

Load Position (cm)	Resistance (ohm)	Inductance (nH)	Capacitance (pF)
25	1600	11	2.2
30	1800	8	2
40	500	50	6.5
65	1300	60	4.5
100	100	375	51
155	300	210	5
170	200	375	5

**Table 2 micromachines-15-01477-t002:** Parameters of optimized load of upper monopole.

Load Position (cm)	Resistance (ohm)	Inductance (nH)	Capacitance (pF)
4.3	1200	2.2	1.35
5.2	1700	2.5	1.42
7.1	1500	25	1.5

**Table 3 micromachines-15-01477-t003:** Comparison of presented works with other reported wide-band monopoles.

Ref.	Operating Band	Gain (dB)	VSWR
[33]	30–450 MHz	>−2	<3.5
[35]	2–30 MHz	>−4.5	<3.5
[39]	30–600 MHz 820–1200 MHz	>−21 >−5	<3 <2
[38]	120–520 MHz	>−4	<2
[40]	3–5 GHz 6–14 GHz	>3 >3	<2 <2
This work	30–750 MHz 800–3000 MHz	>−2.5 >0	<3.5 <2.5

## Data Availability

The data are available upon reasonable request from the corresponding author.

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
