# Peer review of "Design of Two Compact Wideband Monopoles Through Loading with Linear Approximated Lumped Components"

_micromachines, 2024, doi:10.3390/mi15121477_

Round 1
Reviewer 1 Report
Comments and Suggestions for Authors
The authors should explain and supplement the following questions:
1) In Figure 2, the letter (Frequency) of the first word should be a capital case.
2) The calculated or simulated radiation efficiency should be given in the paper.
3) There are many grammatical errors in the full paper that the author should check carefully.
4) How to decide the size of the two monopoles in this article?
5) What is the impact of the combined antenna structure on the radiation performance of the high-frequency antenna?
6) The article lacks detailed instructions on the optimal design of the antenna loading position, so please add more relevant descriptions.
Author Response
Comments 1: In Figure 2, the letter (Frequency) of the first word should be a capital case.
Response 1: Thanks for your comment. We have revised the first word in Figure 2, Figure 5, Figure 6, Figure 7, Figure 12 and Figure 13.
Comments 2: The calculated or simulated radiation efficiency should be given in the paper.
Response 2: Thanks for your advice. We have included discussion of radiation efficiency for the two monopoles in the revised manuscript and this could be found in page 10, section 3, line 304-307 and in page 12, Figure 14.
Comments 3: There are many grammatical errors in the full paper that the author should check carefully.
Response 3: Thanks for your comment. We have revised many words break due to inappropriate line feed.
Comments 4: How to decide the size of the two monopoles in this article?
Response 4: Thanks for your question. The size of low-frequency antenna is decided in consideration of matching difficulty at the lower side of frequency band. The size of high-frequency antenna is decided as long as the size of cross blade is greater than the size of high-frequency antenna so that an elevated monopole is built.
Comments 5: What is the impact of the combined antenna structure on the radiation performance of the high-frequency antenna?
Response 5: Thanks for your question. The radiation performance of an elevated unloaded monopole in the combined antenna structure differs from ordinary unloaded monopole on ground alone. By employing load technique, the performant of the high-frequency antenna has been improved.
Comments 6: The article lacks detailed instructions on the optimal design of the antenna loading position, so please add more relevant descriptions.
Response 6: Thanks for your question. We have revised our paper and discuss this question in page 7, section 2.3, line 235-237.
Reviewer 2 Report
Comments and Suggestions for Authors
Quite interesting paper.
Several issues should be discussed:
- sensitivity analysis of the lumped elements can be added (e.g. consider tolerances)
- what power the antenna can handle? There also could be high circulating currents and high voltages in L, C elements.
- the title is bit misleading: Term collinear is usually used in the area of high gain collinear structures (like Franklin antenna, Bruce array etc.)
- radiation patterns at several frequencies should be included (at least simulated)
- radiation efficiency should be discussed
Author Response
Comments 1: sensitivity analysis of the lumped elements can be added (e.g. consider tolerances)
Response 1: Thanks for your advice. Here we focus on introducing a method about loading with approximated lumped model and design of two compact loaded monopole. Sensitivity analysis for each element of each load is another big topic. Since there are thirty elements in ten loads of the two monopoles.
Comments 2: what power the antenna can handle? There also could be high circulating currents and high voltages in L, C elements.
Response 2: Thanks for your good question. In our load, inductance is winded by wire with diameter of 2.5mm and could handle high power. Loaded capacitance is carried out by ATC capacitance which could handle high voltages up to 500V.
Comments 3: the title is bit misleading: Term collinear is usually used in the area of high gain collinear structures (like Franklin antenna, Bruce array etc.)
Response 3: Thanks very much for good proposal. We want to express some proposed shared structures in the two monopoles which could result in compact design. And We have made a revise in title.
Comments 4: radiation patterns at several frequencies should be included (at least simulated) Response 4: Thanks for your advice. We have included discussion of radiation patterns for the two monopoles at several frequencies in the revised manuscript and this could be found in page 8, section 2.3, line 262-266 and in page 9, Figure 8 and page 10, Figure 9.
Comments 5: radiation efficiency should be discussed
Response 5: Thanks for your advice. We have included discussion of radiation efficiency for the two monopoles in the revised manuscript and this could be found in page 10, section 3, line 304-307 and in page 12, Figure 14.
Round 2
Reviewer 2 Report
Comments and Suggestions for Authors
Thanks to authors for taking into account my comments and addressing them in the new version.